green chemistry/energy

palm fatty acid distillate methyl ester, response surface methodology, methyl ester-petro diesel blends, FTIR, TGA

**Author for correspondence:**
Umer Rashid
e-mail: umer.rashid@upm.edu.my;
umer.rashid@yahoo.com

This article has been edited by the Royal Society of Chemistry, including the commissioning, peer review process and editorial aspects up to the point of acceptance.

# Optimization and blends study of heterogeneous acid catalyst-assisted esterification of palm oil industry by-product for biodiesel production

Shehu-Ibrahim Akinfalabi[1], Umer Rashid[1],
Imededdine Arbi Nehdi[3,4], Thomas Shean Yaw Choong[2],
Hassen Mohamed Sbihi[3] and Mohamed Mossad Gewik[3]

[1]Institute of Advanced Technology, and [2]Department of Chemical and Environmental Engineering, Engineering Faculty, Universiti Putra Malaysia, 43400 UPM Serdang, Selangor, Malaysia
[3]Department of Chemistry, College of Science, King Saud University, Riyadh 11451, Saudi Arabia
[4]Chemistry Department, El Manar Preparatory Institute for Engineering Studies, Tunis El Manar University, P.O. Box 244, Tunis 2092, Tunisia

UR, 0000-0001-6224-413X

The optimum conditions to produce palm fatty acid distillate (PFAD)-derived-methyl esters via esterification have been demonstrated with the aid of the response surface methodology (RSM) with central composite rotatable design in the presence of heterogeneous acid catalyst. The effect of four reaction variables, reaction time (30–110 min), reaction temperature (30–70°C), catalyst concentration (1–3 wt.%) and methanol : PFAD molar ratio (3 : 1–11 : 1), were investigated. The reaction time had the most influence on the yield response, while the interaction between the reaction time and the catalyst concentration, with an $F$-value of 95.61, contributed the most to the esterification reaction. The model had an $R^2$-value of 0.9855, suggesting a fit model, which gave a maximum yield of 95%. The fuel properties of produced PFAD methyl ester were appraised based on the acid value, iodine value, cloud and pour points, flash point, kinematic viscosity, density, ash and water contents and were compared with biodiesel EN 14214 and ASTM D-6751 standard limits. The PFAD methyl ester was further blended with petro-diesel from B0, B3, B5, B10, B20 and B100, on a volumetric basis.

The blends were characterized by TGA, DTG and FTIR. With an acid value of 0.42 (mg KOH $g^{-1}$), iodine value of 63 (g.$I_2$/100 g), kinematic viscosity of 4.31 (mm$^2$ s$^{-1}$), the PFAD methyl ester has shown good fuel potential, as all of its fuel properties were within the permissible international standards for biodiesel.

# 1. Introduction

Several uncertainties, such as fluctuations in prices, environmental degradation and sustainability surrounding fossil fuel usage and production, have made it extremely important to exploit other viable alternatives. Biodiesel, a renewable and environmental benign fuel, has been proposed as the best alternative source to petroleum fuel [1]. Diesel fuels, produced primarily from vegetable oil and animal fats, are generally categorized as biodiesel, since they are composed of compounds of glycerides and fatty acid alkyl esters [2].

Esterification of fats with high FFA and transesterification of vegetable and waste cooking oils are some of the ways through which biodiesel have been produced [3,4]. Esterification reaction involves the reaction of short-chain alcohol, mostly methanol and fatty acids with high FFA producing FAME with water as the by-product in the presence of an acidic or basic catalyst either homogeneous or heterogeneous [5]. This reaction is mostly influenced by four factors, namely: reaction time, reaction temperature, catalyst concentration and molar ratio of the feedstock and alcohol used [6]. The advantages of biodiesel far outweigh its disadvantages, as biodiesel production hopes to open new ways that will ensure a cleaner and safer environment.

Presently, there are numerous vegetable oil grade feedstocks explored for the production of biodiesel such as rapeseed [7], canola [8], oil palm [9], sunflower [10], etc. However, it is important to note that food-grade vegetable oils have been highly discouraged to avert the food versus fuel debate and shortage in the food supply that continuous usage of vegetable oils will imminently lead to [11]. Some of the non-edible sources explored for biodiesel are: PFAD [3], Jatropha [12], Karanja oil [13], rubber seed oil [14], jojoba [15], neem oil [16] and microalgae [17]. Palm fatty acid distillate (PFAD), a by-product from the palm oil industry, still remains one of the best alternatives as it has a high fatty acid content between 85 and 93% [3,6,18].

A large amount of PFAD is produced during the stripping, deodorization and refining of palm oil. In 2017, Malaysia produced 21 000 MT of palm oil, which accounts for 32% of the global production as the second largest producer of oil palm [19]. This goes to show the abundance of PFAD within the region. PFAD primarily consists of about 90.01% free fatty acid, 9% triglycerides and other small contents of diglycerides, monoglycerides, sterols, vitamin E and other impurities [20].

Furthermore, one of the major challenges in the biodiesel industry is the cost of production due to the expensive synthesis process and feedstock used. Therefore, biodiesel blends (petro-biodiesel mixture) have been proposed as a viable solution that can greatly reduce the cost of fuels, while also reducing the hazardous environmental impacts of petrol fuel [21]. From previous studies, RSM has been used to optimize the esterification reaction of Jatropha oil [22], while Ghadge and Raheman used a similar method to optimize the conditions to reduce the FFA contents of *Madhuca indica* seed oil [23]. Also, similar work has been done where RSM was applied in the process optimization for the transesterification reaction of *Raphanus sativus* seed oil [24].

To the best of our knowledge, no previous work has been done in the process optimization for the esterification of PFAD in the presence of biomass-based heterogeneous catalyst using RSM as an optimization tool. Therefore, the objective of this work is to use response surface methodology central composite rotational design (RSM-CCRD) software for process optimization for the production of FAME from PFAD, while also comparing the results with the blends of FAME-petro-diesel. Four esterification variables have been optimized on five different levels from −2 to +2. The reaction time (h), reaction temperature (°C), catalyst concentration (wt%) and molar ratio of PFAD to methanol were the esterification variables optimized. The synthesized FAME was further characterized using FTIR, DTG and TGA.

# 2. Material and methods

## 2.1. Materials and reagents

We obtained PFAD and the palm seed cake for bio-based catalysts was sourced from Sime Darby Group Sdn. Bhd., Malaysia. All chemicals (reagents and solvents) used in this study were analytical grade and

obtained from Thermo Fisher Scientific International Inc. while Sigma Aldrich supplied all the internal reference standards for fatty acid methyl esters.

## 2.2. Catalyst synthesis

Synthesized palm waste biochar (PWB-$H_2SO_4$) from our previous work [25] was used as the catalyst for this work. Palm waste biochar (PWB) was sulfonated with sulfuric acid after pretreatment with $H_3PO_4$. It exhibited excellent catalytic property such as 11.35 mm $g^{-1}$ acid density, pore diameter of 6.25 nm and surface area of 372.01 $m^2 g^{-1}$.

## 2.3. Characterization of PFAD

The PFAD was characterized by evaluating its physico-chemical properties via the free fatty acid (FFA) content, saponification value (SV), iodine value (IV), water content (WC), acid value (AV) and conventional mass per volume at 50°C (litre weight in air). The AOCS Ca 5a-40 standard method (titration) was used for the evaluation of the acid value and FFA [5]. The AOCS Cd 3–25 method was used for the determination of the saponification value while the moisture content was calculated following the AOCS Aa 3–38 standard method. The conventional mass per volume at 50°C (litre weight in air) was conducted following the ISO 6883 (2017) standard method while the iodine value was determined by the AOCS Cd 1–25 standard method. The various FA compositions (%) of the PFAD were conducted via the GC-FID and NIST 17 mass spectra library was used for the identification of the spectrum of each of the fatty acids.

## 2.4. Design of experimental procedure

The esterification reaction involves the mixture of the desired measurements of PFAD, methanol and bio-based catalyst in a reflux batch flat-bottomed glass reactor. After each run, the mixtures are separated, the FAME is collected and is heated at 100°C for about 60 min to remove excess moisture and methanol.

There are four variables that mostly influence the FFA conversion or FAME yield. They are reaction time, molar ratio of the feedstock to methanol, reaction temperature and catalyst concentration. In this experimental design, we have taken these four parameters and run them in a response surface methodology, and based on the reaction, interaction and properties of the reactants, the factorial levels were selected. The reaction time was varied from 30 to 110 min while the reaction temperature was varied from 30–70°C. The molar ratio between methanol and PFAD was also varied from 3 : 1 to 11 : 1 and the concentration of the catalyst was varied from 1.0 to 3.0 wt.%. The reaction was done in a reflux 500 ml 3-necked, flat bottomed batch glass reactor and immersed in an oil water bath. The response FAME yield was established by the full factorial of the response surface methodology central composite rotatable design (RSM-CCRD). The FAME yield was calculated by [26]:

$$\text{FAME yield } (\%) = \frac{(\Sigma A) - \text{AEI}}{\text{AEI}} \times \frac{\text{CEI} \times \text{VEI}}{m} \times 100, \qquad (2.1)$$

where $Y$ denotes FAME yield (%); $\Sigma A$, total peak area of the FAME (C14:0 to C24:1); AEI, peak area of methyl-heptadecanoate; CEI, concentration of methyl-heptadecanoate solution (mg $l^{-1}$); VEI, volume of methyl-heptadecanoate solution (ml); $m$, mass of the sample (mg).

After each experimental phase, the catalyst is separated from the FAME produced, washed and preserved in an oven for further use. The FAME is weighed and treated with anhydrous sodium sulfate to remove any residual moisture content.

## 2.5. Statistical analysis

The RSM-CCRD, using a full-scale factorial, generated all the interactions between the selected variables. Table 1 shows all the levels of the four selected variables i.e. reaction time ($R_1$), reaction temperature ($R_2$), concentration of catalyst ($R_3$) and molar ratio of methanol : PFAD ($R_4$).

The levels for $R_1$ ranged from 30°C to 110 min while $R_2$ ranged from 30°C to 70°C. $R_3$ was varied from 1.0 to 3.0 wt.% while $R_4$ ranged from 3 : 1 to 11 : 1. The FAME yield response was taken for each of the reaction standard runs against the expected and residual runs as generated from the RSM template. The RSM template also provided the degree of freedom, $p$- and $F$-values. The analysis of variance (ANOVA), which was a tool used to interpret the interaction between all the interaction variables, was also

**Table 1.** Esterification optimization parameters for RSM-CCRD.

| variables | symbol | range and levels | | | | |
|---|---|---|---|---|---|---|
| | | −2 | −1 | 0 | 1 | 2 |
| reaction time (min) | $R_1$ | 30 | 50 | 70 | 90 | 110 |
| reaction temperature (°c) | $R_2$ | 30 | 40 | 50 | 60 | 70 |
| concentration of catalyst (wt.%) | $R_3$ | 1 | 1.5 | 2 | 2.5 | 3 |
| methanol : PFAD molar ratio | $R_4$ | 3 : 1 | 5 : 1 | 7 : 1 | 9 : 1 | 11 : 1 |

evaluated. The RSM template also suggested a quadratic model. The lack of fit, regression model and significance of terms are mainly test parameters used to evaluate and assess the significance and reliability of a model. The most significant variable is observed with respect to its significance of terms which is also further supported by the probable values of $p$ and $f$. Therefore, the higher the $F$-value and the lower the $p$-value of a variable, the more significant the expected effect on the yield response. The regression model ($R^2$) has a range from 0 to 1, and the closer the value is to 1, the better the model is expected to be. Also, a $p$-value of less than 0.005 shows a significant model. A three-dimensional graph is also generated from the RSM-CCRD to show the interaction between all the variables and their respective roles in the FAME yield response.

## 2.6. FAME analysis

To calculate the FAME yield, the EN 14103 method was employed [26] while using the Agilent GC capillary column (0.32 mm × 30 m, 0.25 µm). The temperature of the detector is set at 250°C, injection volume is 1 µl while the oven is set at 210°C. The method employed here requires several internal standards to be prepared (palmitic, oleic, stearic, linolenic, lauric, linoleic and myristic esters) and methyl heptadecanoic acid was prepared as the reference standard [26]. All reference standards, i.e. methyl oleate, methyl palmitate, methyl linoleate, methyl myristate and methyl stearate, were diluted to become 1000 ppm [6]. On average, we injected 1 µl of the sample into the injector pot of the GC while the inlet temperature was set at 250°C and the FID set at 210°C. The GC oven's starting temperature was programmed to start at room temperature and increase to 250°C with the temperature rate set at 10°C min$^{-1}$.

## 2.7. FAME-blend procedure

Currently, there is a global demand from users of biodiesel to always blend it based on the volumetric percentage that best suits their usage. For this study, we have six FAME petro-diesel blends; B0, B3, B5, B10, B20 and B100. Where B0 represents pure petroleum diesel, B3 represents 3% of biodiesel and 97% of petroleum diesel. B5 represents 5% biodiesel while B10 and B20 represent 10% and 20% biodiesel, respectively, and B100 represents 100% biodiesel. Typically, we took 0.15 ml of biodiesel plus 4.85 ml of petroleum diesel to represent B3 while for B20, 1 ml of biodiesel was balanced with 4 ml of petroleum diesel. These blends were then characterized with FTIR and TGA and the following fuel properties were analysed; acid value, cloud point, pour point, flash point, density and kinematic viscosity with their respective standard methods.

## 2.8. FAME-blend physico-chemical characterization

The thermos-gravimetric analysis (TGA) and differential thermogravimetry (DTG) were carried out with the Mettler Toledo 990 instrument in order to investigate the thermal stability of the synthesized FAME. The FTIR spectra were obtained with the aid of the Perkin Elmer 1725X machine. The wavenumber ranged from 500 to 4000 cm$^{-1}$ with a resolution set at 2 cm$^{-1}$.

## 2.9. FAME-blend fuel properties

The kinematic viscosity of the FAME blends was done according to the ASTM D-445 standard method where a homogeneous temperature is maintained within the kinematic viscosity bath. The acid value was

**Table 2.** BET analysis, acid density, FFA conversion and FAME yield of used-PWB and PWB-H$_2$SO$_4$.

| sample | $S_{BET}$ (m$^2$ g$^{-1}$)[a] | $V_p$ (cm$^3$ g$^{-1}$)[a] | $D_p$ (nm)[a] | NH$_3$ acid density (mmol g$^{-1}$)[b] | FFA conversion (%)[c] | FAME Yield (%)[4] | sulfur content (%)[d] |
|---|---|---|---|---|---|---|---|
| PWB-H$_2$SO$_4$ | 372.01 | 0.73 | 6.25 | 11.35 | 97.4 | 95.2 | 1.81 |
| used-PWB-H$_2$SO$_4$ | 335.45 | 0.49 | 6.82 | 0.31 | — | — | 0.02 |

[a]Measured by BET analysis.
[b]Measured byTPD-NH$_3$ analysis.
[c]Measured by GC-FID analysis.
[d]Measured by CHNS elementally analysis.

determined using the ASTM D-664 standard method. The pour point and cloud point were determined using the ASTM D-97 and ASTM D-2500, respectively. The flash point was determined using the ASTM D-93 standard method while the density is determined using the ASTM D-1298 standard method.

# 3. Results and discussion

## 3.1. Catalysts stability

The PWB-H$_2$SO$_4$ catalyst was used for eight reaction runs and was stable through to the fifth successive runs. The first run gave FFA conversion of 97.4% and FAME yield of 96.1%. The second run recorded an FFA conversion of 95.3% and FAME yield of 93.1%. The synthesized catalyst was relatively stable through the fifth run, where the FFA conversion recorded was 82.1% and FAME yield was 76.3%. The reaction was stopped at the ninth reaction run when the conversion and FAME yield drastically dropped.

The recovered catalyst (used-PWB-H$_2$SO$_4$) was investigated with FTIR to confirm the decomposition of the sulfonic group attached to the PWB-H$_2$SO$_4$ catalyst. It is observed that sulfonic group degraded drastically after the eighth reaction run, which can be further reactivated via sulfonation for more runs. As presented in table 2, the synthesized catalyst has a pore diameter of 6.25 nm, pore volume of 0.73 and specific surface area of 372.01 m$^2$ g$^{-1}$, while the used-PWB-H$_2$SO$_4$ recorded 335.45 m$^2$ g$^{-1}$ for specific surface area, 0.49 cm$^3$ gm$^{-1}$ for pore volume, 6.82 nm for pore diameter. It was observed that the surface area and pore diameter of the used-PWB-H$_2$SO$_4$ catalyst was decreased as compared to the fresh PWB-H$_2$SO$_4$ catalyst.

Table 2 shows the acid density of the used-PWB-H$_2$SO$_4$ (0.31 mmol g$^{-1}$) was much less than that of the fresh PWB-H$_2$SO$_4$ catalyst (11.35 mmol g$^{-1}$), which means the acidity and activity of the used catalyst significantly reduces after being used in the esterification reaction. Also, the decrease in the catalyst activity was authenticated by determining the sulfur content. It was observed that the fresh catalyst showed sulfur content of 1.81%, whereas the spent catalyst depicted 0.02% sulfur content after the eighth reaction cycle.

## 3.2. Characterization parameters for PFAD biodiesel

The characteristic analysis of the PFAD feedstock used for the esterification reaction has been presented in table 3. With the AOCS method (Ca 5a–40), the FFA content was calculated as 90 wt.% while the saponification value was 210.15 mg KOH g$^{-1}$ with the AOCS Cd 3–35 standard method as a guide. The AOCS Cd 1–25 standard method was used to calculate the iodine value and we recorded 51.07 gI$_2$/100 g while the water content was 0.09 wt.%. The PFAD conventional mass per volume at 50°C was 0.75 kg l$^{-1}$ using the ISO 6883 standard method, while the molecular weight was 270 g mol$^{-1}$. The acid value recorded was 180 mg KOH g$^{-1}$ using the AOCS Cd 3d-63.

Of the 90 wt.% of FFA contained in the PFAD, palmitic acid (49.23%) and oleic acid (37.91%) were more predominant, consisting of about 87.14% in summation. Linoleic acid 7.87%, stearic acid 3.75% and 1.04% myristic acid were the remaining composition. The remaining 10% comprises about 6% triglycerides (TG), 3% diglycerides (DG), 0.5% monoglycerides (MG) and a number of impurities. The PFAD methyl esters content, on the other hand (table 4), were palmitic acid methyl ester (54.02%),

**Table 3.** PFAD characteristics and standard methods.

| properties | unit | wt.% | methods |
|---|---|---|---|
| saponification value | (mg KOH).(g sample)$^{-1}$ | 210.15 | AOCS Cd 3–25 |
| iodine value | (g.I$_2$).(100 g)$^{-1}$ | 51.07 | AOCS Cd 1–25 |
| free fatty acid content | (wt.%) | 90 | AOCS Ca 5a-40 |
| acid value | (mg KOH).(g sample)$^{-1}$ | 180 | AOCS Cd 3d-63 |
| water content | (%) | 0.09 | AOCS Aa 3–38 |
| conventional mass per volume at 50°C | (kg.I$^{-1}$) | 0.75 | ISO 6883 |
| molecular weight | g mol$^{-1}$ | 270 | — |

**Table 4.** PFAD methyl esters composition.

| PFAD methyl esters | (wt.%) |
|---|---|
| palmitic acid methyl ester | 54.02 |
| stearic acid methyl ester | 2.84 |
| myristic acid methyl ester | 1.44 |
| oleic acid methyl ester | 33.43 |
| linoleic acid methyl ester | 8.12 |
| others | 0.15 |

oleic acid methyl ester (33.40%), stearic acid methyl ester (2.84%), myristic acid methyl ester (1.44%) and linoleic acid methyl ester (8.12%). Other forms of unidentified methyl esters accounted for 0.15%.

## 3.3. Optimum reaction conditions by response surface methodology

In table 5, we have presented the interaction between the PFAD feedstock and all the esterification reaction variables by using the response surface methodology central composite rotational design (RSM-CCRD). Four reaction parameters were selected, i.e. reaction time ($R_1$), reaction temperature ($R_2$), concentration of catalyst ($R_3$) and methanol : PFAD molar ratio ($R_4$). We took levels from the lowest range of −2 to the highest range of +2. The $R_1$ was sampled from 30 to 110 min, $R_2$ 30–70°C, $R_3$ 1–3 wt.% and $R_4$ from 3 : 1 to 11 : 1. With these parameters, the RSM-CCRD generated a template for each reaction run. The achieved yield is recorded as the actual yield response alongside the predicted and residual values. We saw a variation between the lowest and highest yield from 58 to 95%.

The lowest FAME yield (58%) was observed at 50 min, 40°C reaction temperature, 1.5 wt.% catalyst concentration and 5 : 1 methanol : PFAD molar ratio. Whereas, the highest FAME yield was recorded at 90 min, 60°C reaction temperature, 1.5 wt.% catalyst concentration and 9 : 1 methanol : PFAD molar ratio. The design method generated series of equations and suggested the quadratic model as the most suitable for this experimental run. Thus, the quadratic equation is presented below as equation (3.1) in terms of coded factors:

$$Y = 90.16667 + 14.83333 \times R_1 + 5.5 \times R_2 \times 7.83 \times R_3 + 5.5 \times R_4 + 1 \times R_1 R_2 - 18 \times R_1 R_3 - 5.5$$
$$\times R_1 R_4 - 6.5 \times R_2 R_3 + 9 \times R_2 R_4 - R_3 R_4 - 11.08 \times R_1^2 - 2.58 \times R_2^2 - 20.28 \times R_3^2 - 8.08 \times R_4^2, \quad (3.1)$$

where $Y$ stands for the response variable (FAME yield), while $R_1$, $R_2$, $R_3$ and $R_4$ represent the actual reaction variables, i.e. reaction time, reaction temperature, catalyst concentration and methanol : PFAD molar ratio, respectively.

The ANOVA quadratic model, as well as the sequential model sum of squares, has been presented in table 6. The quadratic model has a value of 3457.47 as the sum of squares, a degree of freedom of 14, a mean square of 246.96, an $F$-value of 72.87 and a $p$-value of less than 0.0001, which all contributed to the significance of this model. Also important is the lack of fit, a crucial indicator of a good model. Table 6 shows our model has an insignificant lack of fit.

**Table 5.** Esterification parameters for CCD.

| run order | $R_1$: reaction time (min) | $R_2$: reaction temperature (°C) | $R_3$: catalyst concentration (wt.%) | $R_4$: methanol: PFAD molar ratio | FAME yield (%) | predicted value (%) | residual value (%) |
|---|---|---|---|---|---|---|---|
| 1 | 50 | 40 | 1.5 | 5 : 1 | 58 | 56.12 | 1.8 |
| 2 | 50 | 40 | 1.5 | 9 : 1 | 65 | 63.50 | 1.5 |
| 3 | 50 | 40 | 2.5 | 5 : 1 | 80 | 78.83 | 1.17 |
| 4 | 50 | 40 | 2.5 | 9 : 1 | 79 | 79.95 | −0.95 |
| 5 | 50 | 60 | 1.5 | 5 : 1 | 59 | 60.00 | −1.00 |
| 6 | 50 | 60 | 1.5 | 9 : 1 | 77 | 75.63 | 1.37 |
| 7 | 50 | 60 | 2.5 | 5 : 1 | 78 | 76.96 | 1.04 |
| 8 | 50 | 60 | 2.5 | 9 : 1 | 85 | 86.33 | −1.33 |
| 9 | 90 | 40 | 1.5 | 5 : 1 | 84 | 82.33 | 1.67 |
| 10 | 90 | 40 | 1.5 | 9 : 1 | 83 | 83.46 | −0.46 |
| 11 | 90 | 40 | 2.5 | 5 : 1 | 87 | 87.79 | −0.79 |
| 12 | 90 | 40 | 2.5 | 9 : 1 | 84 | 82.67 | 1.33 |
| 13 | 90 | 60 | 1.5 | 5 : 1 | 88 | 86.46 | 1.54 |
| 14 | 90 | 60 | 1.5 | 9 : 1 | 95 | 95.83 | −0.83 |
| 15 | 90 | 60 | 2.5 | 5 : 1 | 85 | 86.17 | −1.17 |
| 16 | 90 | 60 | 2.5 | 9 : 1 | 88 | 89.29 | −1.29 |
| 17 | 70 | 50 | 2 | 3 : 1 | 75 | 76.70 | −1.70 |
| 18 | 70 | 50 | 2 | 11 : 1 | 88 | 87.21 | 0.79 |
| 19 | 70 | 50 | 1 | 7 : 1 | 59 | 61.38 | −2.38 |
| 20 | 70 | 50 | 3 | 7 : 1 | 79 | 77.54 | 1.49 |
| 21 | 70 | 30 | 2 | 7 : 1 | 80 | 82.22 | −2.2 |
| 22 | 70 | 70 | 2 | 7 : 1 | 94 | 92.71 | 1.29 |
| 23 | 30 | 50 | 2 | 7 : 1 | 63 | 64.38 | −1.38 |
| 24 | 110 | 50 | 2 | 7 : 1 | 94 | 93.54 | 0.46 |
| 25 | 70 | 50 | 2 | 7 : 1 | 89 | 90.16 | −1.16 |
| 26 | 70 | 50 | 2 | 7 : 1 | 90 | 90.16 | −0.16 |
| 27 | 70 | 50 | 2 | 7 : 1 | 91 | 90.17 | 0.83 |
| 28 | 70 | 50 | 2 | 7 : 1 | 92 | 90.17 | 1.83 |
| 29 | 70 | 50 | 2 | 7 : 1 | 89 | 90.17 | −1.17 |
| 30 | 70 | 50 | 2 | 7 : 1 | 90 | 90.17 | −0.17 |

The confidence level at 95% showed an $F$-value of 72.87 while also having a $p$-value of less than 0.0001, showing a significantly fitted model in which the regression analysis can reliably predict the yield response [27]. Furthermore, each of the items addressed in the model at 95% confidence level were found to be significant as seen in their respective $F$-values. These tests have shown that the model is satisfactory in the prediction of the yield response with respect to all the variables studied and also highlight the validity of the quadratic model as a regressional tool for this study. On the other hand, $p$-values that are below 0.0001 show a strong indication of a significant model [28].

Moreover, the individual variables have been indicated to be more influential in the yield response than their collective interactions as evidenced by their corresponding very low $p$-values and higher $F$-values. The $F$-value recorded for the reaction time ($R_1$) is 389.56, which according to this model has the most influence on the reaction yield response.

**Table 6.** ANOVA quadratic model for response surface.

| source | sum of squares | d.f. | mean square | $F$-value | $p$-value | |
|---|---|---|---|---|---|---|
| model | 3457.47 | 14 | 246.96 | 72.87 | <0.0001 | significant |
| $R_1$ | 1320.17 | 1 | 1320.17 | 389.56 | <0.0001 | |
| $R_2$ | 181.50 | 1 | 181.50 | 53.56 | <0.0001 | |
| $R_3$ | 368.17 | 1 | 368.17 | 108.64 | <0.0001 | |
| $R_4$ | 181.50 | 1 | 181.50 | 53.56 | <0.0001 | |
| $R_1R_2$ | 1.0000 | 1 | 1.0000 | 0.2951 | 0.5950 | |
| $R_1R_3$ | 324.00 | 1 | 324.00 | 95.61 | <0.0001 | |
| $R_1R_4$ | 30.25 | 1 | 30.25 | 8.93 | 0.0092 | |
| $R_2R_3$ | 42.25 | 1 | 42.25 | 12.47 | 0.0030 | |
| $R_2R_4$ | 81.00 | 1 | 81.00 | 23.90 | 0.0002 | |
| $R_3R_4$ | 49.00 | 1 | 49.00 | 14.46 | 0.0017 | |
| $R_1^2$ | 210.58 | 1 | 210.58 | 62.14 | <0.0001 | |
| $R_2^2$ | 11.44 | 1 | 11.44 | 3.38 | 0.0860 | |
| $R_3^2$ | 726.30 | 1 | 726.30 | 214.32 | <0.0001 | |
| $R_4^2$ | 112.01 | 1 | 112.01 | 33.05 | <0.0001 | |
| residual | 50.83 | 15 | 3.39 | | | |
| lack of fit | 44.00 | 10 | 4.40 | 3.22 | 0.1044 | not significant |
| pure error | 6.83 | 5 | 1.37 | | | |
| cor total | 3508.30 | 29 | | | | |

The second most influential is $R_3$, the catalyst concentration, which has an $F$-value of 108.64 and a $p$-value of less than 0.0001. $R_2$ and $R_4$, the reaction temperature and methanol : PFAD molar ratio, respectively, both recorded the same $p$-value (less than 0.0001) and $F$-value (53.56). The interactions between the models, especially $R_1R_3$, were the most influential as it has the highest $F$-value at 95.61, while also maintaining a $p$-value that is less than 0.0001.

Also important to note is the determination coefficient and regression equation ($R^2$). An $R^2$-value of close to 1 is a strong indicator of a fitted model which can be used for an accurate prediction. The derived $R^2$-value of 0.9855 from our response yield indicates a high precision in the prediction of the model [29]. In support of the predicted value against the actual FAME yield response is figure 1, which presented the plots between the predicted and actual values. The adjusted determinant coefficient ($R_{adj}^2 = 0.9720$) and the actual determination coefficient ($R^2 = 0.9855$) are very close which shows a very close relation and relevance of the model. The observed coefficient of variance (C.V = 2.25%) also indicated near-perfect and reliable results from the model [30].

Figure 2 presents the plot of normal probability which indicated that the errors are evenly distributed along a straight line and insignificant, which again points to a well-fitted model [30]. Table 7 shows the regression coefficients and significance of probability values, showing a suitable model for the representation of the interactions among all the selected variables.

Figure 3$a$–$f$ presents the three-dimensional model graph surfaces of all the interactions among the variables. The curvature of these graphs shows the extent to which they interact and what variable has the most influence in the response yield. Figure 3$a$ shows the interaction between reaction time ($R_1$) and reaction temperature ($R_2$) while figure 3$b$ presents the interaction of reaction time ($R_1$) and catalyst concentration ($R_3$).

Of all the interactions, the highest recorded is between $R_1$ and $R_3$, which also shows the highest $F$-value (95.61) among all the interaction while the least important interaction recorded in terms of the $F$-values is between $R_1$ and $R_2$ where the $F$-value is 0.2951, therefore it contributed least to the response yield. Hence, the optimum condition for the esterification reaction is 90 min reaction time, 60°C reaction temperature, 1.5 catalyst concentration wt.% and 9 : 1 methanol : PFAD molar ratio.

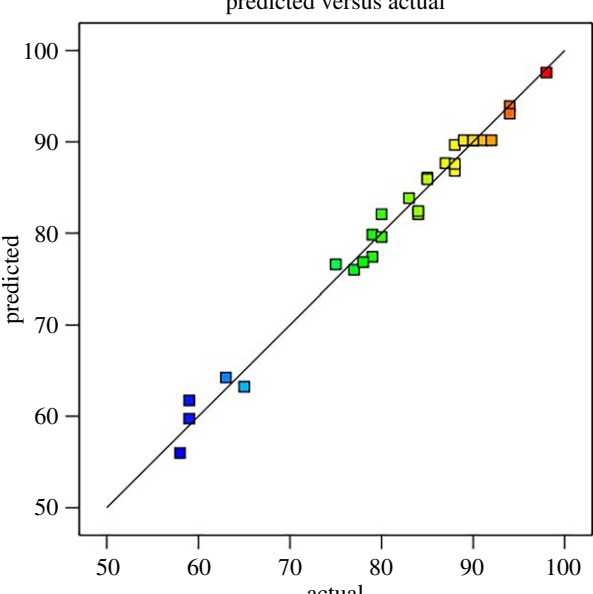

**Figure 1.** Plot of the actual versus predicted values.

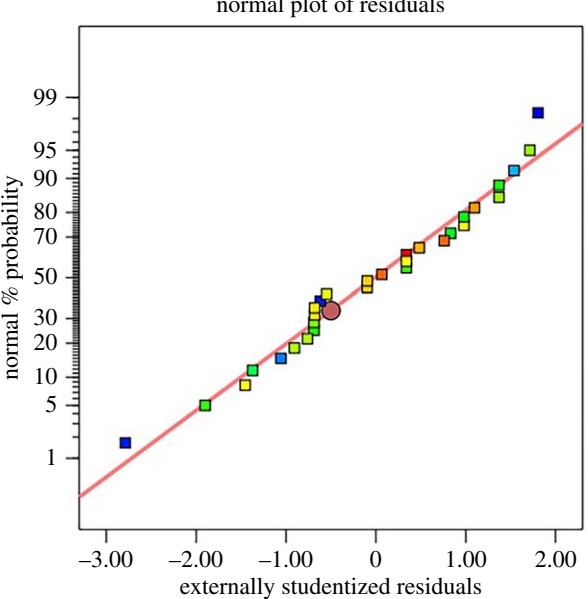

**Figure 2.** Normal probability plot of residuals.

## 3.4. Fuel properties of produced FAME

We have presented the produced PFAD methyl esters composition in table 4. Palmitic acid methyl ester has the highest methyl ester content with 54.02 (wt.%), closely followed by oleic acid methyl ester (33.43 wt.%). The others are linoleic acid methyl ester, stearic acid methyl ester and myristic acid methyl ester at 8.12, 2.84 and 1.44 (wt.%), respectively. Table 8 contains the fuel properties of the produced PFAD methyl ester with biodiesel standards (EN 14214 and ASTM D-6751) comparison as well as the Malaysian petro-diesel.

### 3.4.1. Kinematic viscosity

A key property of the fuel is its kinematic viscosity since it directly affects the operating pump. Higher fuel viscosity makes fuel injection into the fuel collector difficult since the fuel atomization is disturbed.

**Table 7.** Regression coefficients and significance of response surface for quadratic model.

| model terms | coefficient estimate | d.f. | standard error | 95% CI low | 95% CI high | VIF |
|---|---|---|---|---|---|---|
| intercept | 90.17 | 1 | 0.7515 | 88.56 | 91.77 | |
| $R_1$ | 14.83 | 1 | 0.7515 | 13.23 | 16.44 | 1.0000 |
| $R_2$ | 5.50 | 1 | 0.7515 | 3.90 | 7.10 | 1.0000 |
| $R_3$ | 7.83 | 1 | 0.7515 | 6.23 | 9.44 | 1.0000 |
| $R_4$ | 5.50 | 1 | 0.7515 | 3.90 | 7.10 | 1.0000 |
| $R_1R_2$ | 1.00 | 1 | 1.84 | −2.92 | 4.92 | 1.0000 |
| $R_1R_3$ | −18.00 | 1 | 1.84 | −21.92 | −14.08 | 1.0000 |
| $R_1R_4$ | −5.50 | 1 | 1.84 | −9.42 | −1.58 | 1.0000 |
| $R_2R_3$ | −6.50 | 1 | 1.84 | −10.42 | −2.58 | 1.0000 |
| $R_2R_4$ | 9.00 | 1 | 1.84 | 5.08 | 12.92 | 1.0000 |
| $R_3R_4$ | −7.00 | 1 | 1.84 | −10.92 | −3.08 | 1.0000 |
| $R_1^2$ | −11.08 | 1 | 1.41 | −14.08 | −8.09 | 1.05 |
| $R_2^2$ | −2.58 | 1 | 1.41 | −5.58 | 0.4135 | 1.05 |
| $R_3^2$ | −20.58 | 1 | 1.41 | −23.58 | −17.59 | 1.05 |
| $R_4^2$ | −8.08 | 1 | 1.41 | −11.08 | −5.09 | 1.05 |
| s.d. | 1.84 | | $R^2$ | 0.9855 | | |
| mean | 81.70 | | adjusted $R^2$ | 0.9720 | | |
| C.V. % | 2.25 | | predicted $R^2$ | 0.9250 | | |
| | | | adeq precision | 31.9452 | | |

Therefore, petro-diesels are found to have higher kinematic viscosity because of their high molecular weight, increased amount of saturated fatty acids and long carbon chains [32]. Kinematic viscosity of biodiesel can generally be decreased by increasing the catalysis temperature which will bring about thermal cracking of the carbon chains and allows the dehydrogenation reaction [33]. As a result, smaller and weaker molecules allow a reduction in free fatty acid composition in terms of carbon chains [34]. The kinematic viscosity ($mm^2 s^{-1}$; 40°C) shows a little discrepancy between the produced PFAD methyl ester and the standards. The produced methyl ester recorded 4.31, while the Malaysian petro-diesel is 3.8, the EN 14214 standard limit is 3.5–5 $mm^2 s^{-1}$; 40°C while ASTM D-6751 allowable limit is 1.9–6 $mm^2 s^{-1}$; 40°C. This shows that the viscosity of PFAD methyl ester is within the allowable range.

### 3.4.2. Cloud and pour point

The cold flow properties consisting of cloud point CP and pour point PP are the main low-temperature characteristics that define a healthy biodiesel for engines. CP is generally considered as the temperature at which fuel starts to solidify as the temperature drops. These solidified particles can block the plug filter, thereby leading to the reduction in the flow of the fuel. Hence, it is important that this problem is properly taken care of to avoid such a hazard. The temperature at which fluid starts flowing is regarded as the PP. It must be ensured that both the PP and CP are very low so as to prevent freezing [35]. The produced PFAD biodiesel has a CP of 5.5°C and PP of −3°C as presented in table 8. In comparison, the EN 14214 standard limit is −4.

### 3.4.3. Flash point

The temperature at which fuel tends to vaporize so as to produce a combustible mixture in the presence of oxygen is regarded as the flash point FP [36]. Understanding the FP of all fuels is critical in the storage, handling and transportation of fuels. Conversely, a high FP value will be useful in the management of fuels for safety reasons, since the risk of combustion will be seriously reduced. Biodiesel generally

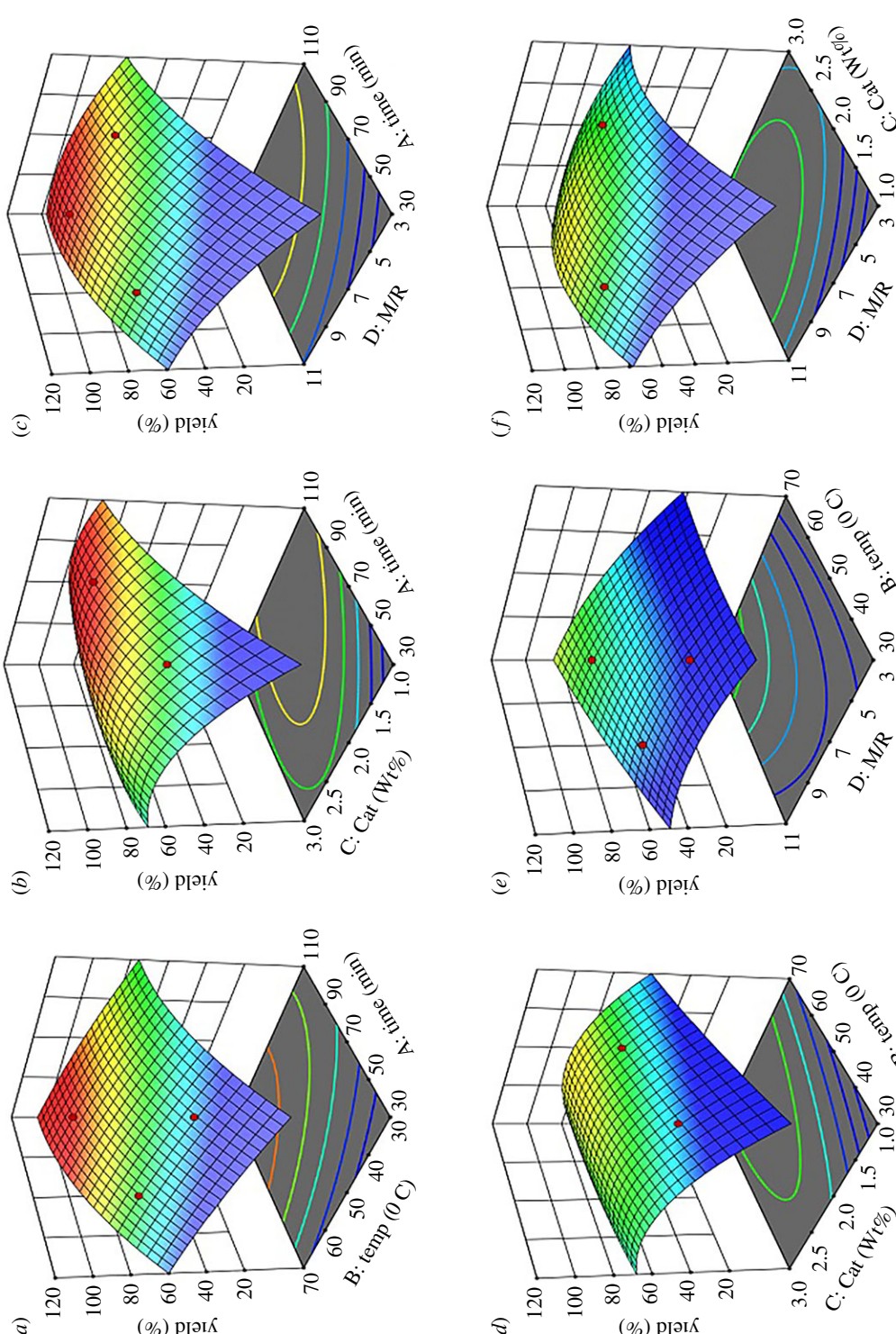

**Figure 3.** Three-dimensional response surface model graphs of interactions of the esterification variables.

**Table 8.** Fuel properties of PFAD methyl esters with biodiesel standards comparison.

| fuel property | PFAD methyl ester | Jatropha-based methyl ester [29] | moringa oil methyl ester [2] | Microalga *Spirulina platensis* methyl ester [31] | Malaysian petro-diesel | EN 14214 | ASTM D6751 |
|---|---|---|---|---|---|---|---|
| kinematic viscosity (mm$^2$ s$^{-1}$, 40°C) | 4.31 | 4.8 | 4.8 | 12.4 | 3.82 | 3.5–5 | 1.9–6 |
| oxidative stability (min) | 4.23 | — | 3.52 | — | — | 3 min | 6 min |
| cloud point (°C) | 5.5 | 10 | 18.0 | −3 | 6 | −4 | — |
| pour point (°C) | −3 | 6 | 17.0 | −9 | −3 | — | — |
| flash point (°C) | 176 | 188 | 162.0 | — | 82 | >101 | >130 |
| density (25°C, kg m$^{-3}$) | 0.872 | 880 | 875 | 864 | 0.870 | 0.86–0.9 | — |
| acid value (mg KOH g$^{-1}$) | 0.42 | 0.40 | 0.38 | 0.75 | 0.25 | 0.50 max | 0.50 max |
| ash content (%) | insignificant | 0.016 | 0.010 | — | 0.01 | 0.02 max | 0.02 max |
| copper strip corrosion (50°C, 3 h) | 1a | 1a | 1a | 1a | 1 | Class 1 | No. 3 max |
| carbon residue (%) | 0.05 | — | — | — | 0.20 | <0.03 | <0.05 |
| sulfur content (%) | 0.01 | 0.011 | 0.0124 | — | 0.14 | 0.05 max | — |
| iodine value (g I$_2$).(100 g)$^{-1}$ | 63 | — | — | 102 | — | <120 | — |
| water content (mg kg$^{-1}$) | 0.01 | — | — | 39 | 0.01 | 0.05 max | 0.03 max |

have lower FP, which is a distinct advantage it has over other fuels [37]. The produced PFAD methyl ester has an FP of 176°C which is within the acceptable limit, while the petro-diesel recorded 82°C. The EN 14214 standards stipulates greater than 101°C while the ASTM D-6751 > 130°C.

### 3.4.4. Acid and iodine value

Acid value is defined as the amount of free acids available in a given sample. This property directly influences the longevity of the fuel [38]. The PFAD biodiesel has an acid value of 0.42 (mg KOH g$^{-1}$). The Malaysian petro-diesel recorded 0.25 and the allowable EN 14214 and ASTM D-6751 limit is 0.50, which shows the produced methyl ester is within the allowable fuel range. The iodine value measures the level of unsaturation in a biodiesel. The methyl ester recorded 63 (g I$_2$/100 g oil) while according to the EN 14214 standard method, the permissible limit is less than 120.

### 3.4.5. Other properties

The oxidative stability (min) was 4.23 min for the PFAD methyl ester, whereas the allowable international limit is 3 min for EN 14214 and 6 min for ASTM D-6751. This ensures that the produced biodiesel meets the requirement for a fuel. One of the most outstanding advantages of the biodiesel over petro-diesel is the sulfur content. The produced PFAD biodiesel is mostly free from sulfur, while petro-diesel contains about 0.14% of sulfur. Therefore, it meets the global requirement and demand for a sulfur-free diesel, since the sulfur oxide that is normally produced in petro-diesel combustion, which led to a couple of environmental problems, has been averted. The density (25°C kg m$^{-3}$) of the PFAD methyl ester is 0.872, while the Malaysian petro-diesel was 0.870 and the EN 14214 standard limit is 0.86–9. This property is important because it affects the ease of atomization of the fuel in an airless combustion [39]. The ash content of the PFAD methyl ester is insignificant while the Malaysian petro-diesel recorded 0.01 wt.%. Although this is still a very small amount and within the permissible limit, it is important to note that ash can be deposited in some parts of the engine and cause abrasion. Bio-based fuels are mostly free of sediments, otherwise they can affect the fuel combustion via abrasion. This may lead to clogged fuel filters and cause shortage in fuel supply to the engine. Water content in fuels is highly undesirable, as they may contribute to microbial growth, hydrolytic oxidation and fuel chamber corrosion [40]. The water content of the PFAD methyl ester is 0.01 (mg kg$^{-1}$), which is lower than the required 0.05 maximum limit for the EN 14214 and 0.03 maximum limit for the ASTM D-6751 standards. The carbon residue recorded for the PFAD methyl ester is 0.05 while that of Malaysian petro-diesel is 0.20. The EN 14214 maximum limit is 0.03 while the ASTM D-6751 standard limit is 0.05 max. This can be said to be an added advantage to the PFAD methyl ester.

### 3.4.6. Comparison of PFAD methyl ester with other produced biodiesel in the literature

As presented in table 8, the PFAD methyl ester (PFADME) has shown good fuel properties as compared to other biodiesel from *Jatropha curcas* methyl ester (JCME) [29], microalgae *spirulina platensis* methyl ester (MSPME) [31] and *Moringa oleifera* methyl ester (MOME) [2]. The kinematic viscosity of synthesized PFADME was 4.31 mm$^2$ s$^{-1}$, which is the lowest value as compared to JCME; 4.8 mm$^2$ s$^{-1}$ [29], MOME; 4.8 mm$^2$ s$^{-1}$ [2] and MSPME; 12.4 mm$^2$ s$^{-1}$ [31]. The acid value for the synthesized PFADME was 0.42 (mg KOH g$^{-1}$), while that of MOME was 0.38 (mg KOH g$^{-1}$), 0.75 (mg KOH g$^{-1}$) for MSPME was 0.40 (mg KOH g$^{-1}$) for JCME. The oxidative stability (min), 4.23 for synthesized PFADME was higher than the 3.52 min MOME. The other fuel properties recorded for the produced PFADME and other biodiesel showed some similarity and even better fuel properties as evidenced in the flash point, cloud and pour point values, as well as copper strip corrosion, carbon residue and iodine value in table 8.

## 3.5. Characterization of PFAD methyl ester blends

Blends from the PFAD methyl ester and petro-diesel have been highlighted via the FTIR analysis and TGA/DTG and presented in figures 4 and 5. Four blends were prepared (B3, B5, B10 and B20) while also maintaining B0 and B100. The blends were prepared as follows: B0 (pure petro-diesel), B3 (3% of PFAD biodiesel and 97% petro-diesel), B5 (5% PFAD biodiesel and 95% petro-diesel), B10 (10% PFAD biodiesel and 90% petro-diesel), B20 (20% PFAD biodiesel and 80% petro-diesel) and B100 (100% biodiesel) on a volumetric basis.

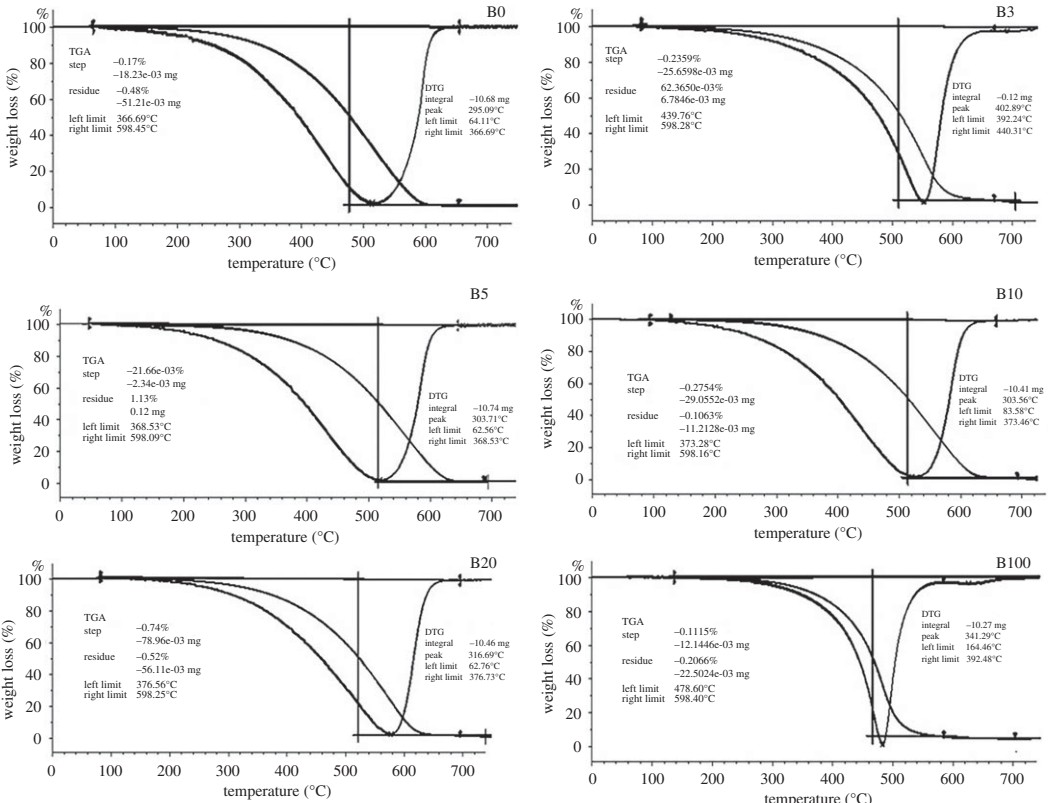

**Figure 4.** TGA and DTG Curves illustration of PFAD biodiesel and petro-diesel blends.

### 3.5.1. TG and DTG analyses

The determination of the thermal strength of the PFAD-derived biodiesel and petro-diesel blends were analysed via the thermogravimetric and derivative thermogravimetric analyses, where the relative weight loss is calculated against the increase in temperature (figure 4). The desorption peaks indicate the temperature at which an active material or functional decomposes. Furthermore, TGA can be effectively used to measure the oxidation stability of produced biodiesel and the left and right degradation limits can also indicate the ability of the synthesized biodiesel to resist thermal degradation [41].

Each exothermic thermal effect marks the onset of a degradation in the DTG curve. The first exothermic thermal event for B0 occurred at approximately 366.69°C as the left limit, while the right limit is at 598.45°C, where the maximum weight loss was recorded. Between the first exothermic event and the last, we observed a steady degradation of the B0. The weight loss recorded at approximately 367°C may be attributable to the volatilization of the pure petro-diesel.

In the first exothermic event, the B3 had a weight loss of about 25%, where the left limit degradation recorded was at 439.76°C and the right limit degradation was at 598.28°C where the decomposition can be attributable to the degradation of the major alkyl group contained in the produced biodiesel. Similar degradation points were observed for B5, B10 and B20, where their onset exothermic events were between 360°C and 380°C for the left limit and they all maintained the same right limit of 598°C. For the DTG, the left limit recorded for the blends were between 62 and 84°C and the right limit of 368–377°C.

These disintegration peaks are evidence of the similarities in the properties of both petro-diesel and biodiesel. Interestingly, we observed that B100 had more thermal resistance than B0. The left limit recorded was 478.60 while the right limit was 598.40, whereas the DTG's left limit peaked at 341.29°C, having a left limit of 164.46°C and right limit of 392.48°C. These degradation peaks for all the blends show thermally stable products even though the standard limit is not specified for thermal stability. Generally, the accepted practice or limit for thermal stability is around 150°C. Hence, any produced biodiesel that is stable above 150°C is regarded as thermally stable. The probable reason why the biodiesel blends have shown good stability may be its resistance to auto oxidation of saturated fatty acid methyl esters [31].

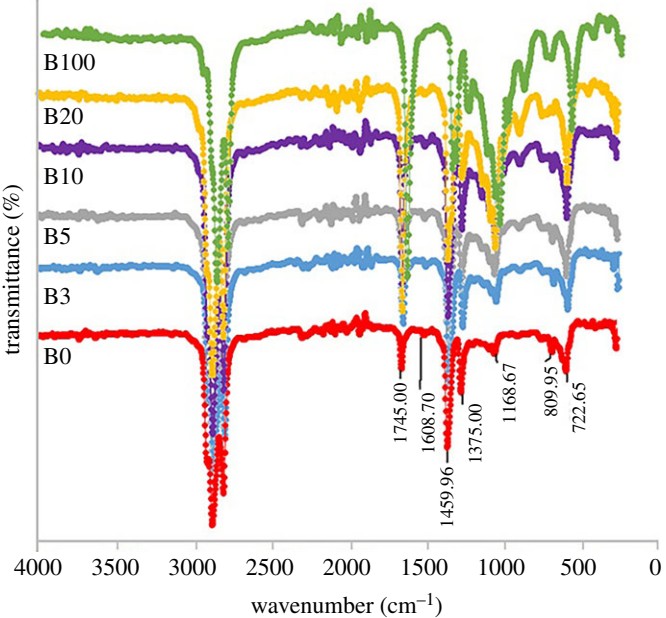

**Figure 5.** FTIR spectra of PFAD biodiesel and petro-diesel blends.

### 3.5.2. FTIR analysis

The derived PFAD-derived biodiesel was blended with Malaysian petro-diesel and characterized using the Fourier Transform Infrared (FTIR) analysis. The FIR spectra are shown in figure 5. From the spectra, we can observe that major components of petroleum diesel are aliphatic hydrocarbons, consisting of long carbon chains in the chemical structure which are similar to biodiesel. The most noticeable adsorption bands for C-H vibrations were observed around 2953.38, 2920.86 and 2852.38 for all the blends from B0 to B100, which corresponds to the symmetric and asymmetric stretching bands which is a characteristic of vibration bands of methyl esters. Also, the adsorption peak recorded at 1256.52 cm$^{-1}$ is an attribute of the asymmetric axial vibration stretching of C(C=O)–O bonds of methyl esters while the stretching bands at 1168.67 cm$^{-1}$ is attributable to the asymmetric axial vibration stretching of O–C–C bonds. Furthermore, for all the blends the C=O vibrational band is observed at 1745.00 cm$^{-1}$, while the O–CH$_3$ band is observed at 1196.99 cm$^{-1}$ and these bands are attributable mainly to mono-alkyl esters [31]. These peaks have shown the immense similarity of all the bands, indicating that the PFAD-derived biodiesel can easily serve as a viable blend with petroleum diesel.

## 4. Conclusion

We have optimized PFAD esterification reaction with RSM-CCRD software, which gave us optimum esterification reaction conditions of 90 min reaction time, 60°C reaction temperature, 1.5 wt.% catalyst concentration and 9 : 1 methanol : PFAD molar ratio. These conditions gave a methyl ester yield of 95%. Of the four reaction variables studied, the reaction time had the most influence on the yield response with an *F*-value of 389.56. The interaction between the reaction time and catalyst concentration contributed the most to the yield response, with an *F*-value of 95.61. The characterization of the PFAD-derived methyl ester was achieved with the FTIR, TGA and DTG analyses. The fuel properties' study of the produced methyl ester showed a thermally stable, sulfur-free and low kinematic viscosity biodiesel. The cold flow properties, flash point, acid and iodine values and density were all within the permissible limits of EN 14214 and ASTM D-6751. The blends between the methyl ester and petro-diesel showed an excellent result as evidenced in the TGA and FTIR analyses. The PFAD has, therefore, exhibited extremely important potential as a fuel feedstock for biodiesel production.

Data accessibility. The datasets supporting this article have been uploaded as part of the electronic supplementary material.
Authors' contributions. Conceptualization, U.R. and S.-I.A.; methodology and software design, U.R.; experimental work, S.-I.A. and M.M.G.; writing—original draft, S.-I.A.; help with editing, supervision and review of the manuscript,

U.R.; scientific guidance and review the final draft of the manuscript, T.S.Y.C.; help to investigate the product and review of the final draft of the paper, I.A.N. and H.M.S.

Competing interests. We declare we have no competing interests.

Funding. This research was funded by Putra IPS grant at the Universiti Putra Malaysia (UPM), through Putra IPS research grant's project number, GP-IPS/2016/9474900. The authors particularly wish to acknowledge the financial support from Universiti Putra Malaysia (UPM) via Putra IPB grant (GP-IPB/2016/9490400). The authors acknowledge their gratitude to King Saud University (Riyadh, Saudi Arabia) for the funding of this research through Researchers Supporting Project number (RSP-2019/80).

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
