## [Reviewer comments · Royal Society Open Science]

Review History

RSOS-191592.R0 (Original submission)

Review form: Reviewer 1

Is the manuscript scientifically sound in its present form?

Yes

Are the interpretations and conclusions justified by the results?

Yes

Is the language acceptable?

Yes

Do you have any ethical concerns with this paper?

No

Have you any concerns about statistical analyses in this paper?

No

Recommendation?

Major revision is needed (please make suggestions in comments)

Comments to the Author(s)

In this submitted manuscript, the authors described the esterification of palm oil industry by-product to produce biodiesel, and studied the properties of the resulted biodiesel. Some useful results were contained. In my opinion, this manuscript could be accepted after considering the following two points.

1. The stability of the used heterogeneous catalyst should be discussed. If possible, the properties of the synthesized catalyst and the recovered catalyst should be characterized.
2. The advantage of the resulted biodiesel should be discussed. At least, a comparison in various properties should be provided between the synthesized biodiesel and some reported typical biodiesel.

Review form: Reviewer 2

Is the manuscript scientifically sound in its present form?

Yes

Are the interpretations and conclusions justified by the results?

Yes

Is the language acceptable?

Yes

Do you have any ethical concerns with this paper?

No

Have you any concerns about statistical analyses in this paper?

No

Recommendation?

Accept with minor revision (please list in comments)

Comments to the Author(s)

The manuscript describes the biodiesel production from palm oil industry by-products by acid-catalyzed esterification reactions. The reaction conditions were thoroughly optimized by the response surface methodology (RSM). The various properties of the produced fuels and the blends with petro-diesel were analyzed. It is recommended to be published after revisions.

1. In Table 1, the first column (variables) can be elongated to accommodate the whole word. Besides, the detailed reaction conditions can be provided as the footnote.
2. In Table 2, the "Wt.%" should be changed into "values", because the unit of the values are not all by weight.
3. The FTIR spectra are shown in an unclear manner. The figure should be replotted with higher resolution and different blends should be denoted by various colors.
4. The recyclability of the heterogeneous catalysts should be examined.

Decision letter (RSOS-191592.R0)

15-Oct-2019

Dear Dr Rashid:

Title: Optimization and blends study of heterogeneous acid catalyst assisted esterification of palm oil industry by-product for biodiesel production
Manuscript ID: RSOS-191592

The editor assigned to your manuscript has now received comments from reviewers. We would like you to revise your paper in accordance with the referee and Subject Editor suggestions which can be found below (not including confidential reports to the Editor). Please note this decision does not guarantee eventual acceptance.

Please submit your revised paper before 07-Nov-2019. Please note that the revision deadline will expire at 00.00am on this date. If we do not hear from you within this time then it will be assumed that the paper has been withdrawn. In exceptional circumstances, extensions may be possible if agreed with the Editorial Office in advance. We do not allow multiple rounds of revision so we urge you to make every effort to fully address all of the comments at this stage. If deemed necessary by the Editors, your manuscript will be sent back to one or more of the original reviewers for assessment. If the original reviewers are not available we may invite new reviewers.

Please also include the following statements alongside the other end statements. As we cannot publish your manuscript without these end statements included, if you feel that a given heading is not relevant to your paper, please nevertheless include the heading and explicitly state that it is not relevant to your work.

- Acknowledgements

Yours sincerely,
Dr Laura Smith

Publishing Editor, Journals

RSC Associate Editor:
Comments to the Author:
(There are no comments.)

RSC Subject Editor:
Comments to the Author:
(There are no comments.)

Reviewers' Comments to Author:
Reviewer: 1

Comments to the Author(s)

In this submitted manuscript, the authors described the esterification of palm oil industry by-product to produce biodiesel, and studied the properties of the resulted biodiesel. Some useful results were contained. In my opinion, this manuscript could be accepted after considering the following two points.

1. The stability of the used heterogeneous catalyst should be discussed. If possible, the properties of the synthesized catalyst and the recovered catalyst should be characterized.
2. The advantage of the resulted biodiesel should be discussed. At least, a comparison in various properties should be provided between the synthesized biodiesel and some reported typical biodiesel.

Reviewer: 2

Comments to the Author(s)

The manuscript describes the biodiesel production from palm oil industry by-products by acid-catalyzed esterification reactions. The reaction conditions were thoroughly optimized by the response surface methodology (RSM). The various properties of the produced fuels and the blends with petro-diesel were analyzed. It is recommended to be published after revisions.

1. In Table 1, the first column (variables) can be elongated to accommodate the whole word. Besides, the detailed reaction conditions can be provided as the footnote.
2. In Table 2, the "Wt.%" should be changed into "values", because the unit of the values are not all by weight.
3. The FTIR spectra are shown in an unclear manner. The figure should be replotted with higher resolution and different blends should be denoted by various colors.
4. The recyclability of the heterogeneous catalysts should be examined.

Author's Response to Decision Letter for (RSOS-191592.R0)

See Appendix A.

RSOS-191592.R1 (Revision)

Review form: Reviewer 1

Is the manuscript scientifically sound in its present form?

Yes

Are the interpretations and conclusions justified by the results?

Yes

Is the language acceptable?

Yes

Do you have any ethical concerns with this paper?

No

Have you any concerns about statistical analyses in this paper?

No

Recommendation?

Accept as is

Comments to the Author(s)

After carefully reviewing the revised version, I am satisfactory with the answers and changes provided by the authors. Therefore, I recommend the revised manuscript to be accepted.

Review form: Reviewer 2

Is the manuscript scientifically sound in its present form?

Yes

Are the interpretations and conclusions justified by the results?

Yes

Is the language acceptable?

Yes

Do you have any ethical concerns with this paper?

No

Have you any concerns about statistical analyses in this paper?

No

Recommendation?

Accept as is

Comments to the Author(s)

The issues have been properly addressed.

Decision letter (RSOS-191592.R1)

19-Nov-2019

Dear Dr Rashid:

Title: Optimization and blends study of heterogeneous acid catalyst assisted esterification of palm oil industry by-product for biodiesel production

Manuscript ID: RSOS-191592.R1

It is a pleasure to accept your manuscript in its current form for publication in Royal Society Open Science. The chemistry content of Royal Society Open Science is published in collaboration with the Royal Society of Chemistry.

RSC Associate Editor:
Comments to the Author:
(There are no comments.)

RSC Subject Editor:
Comments to the Author:
(There are no comments.)

Reviewer(s)' Comments to Author:
Reviewer: 2

Comments to the Author(s)
The issues have been properly addressed.

Reviewer: 1

Comments to the Author(s)
After carefully reviewing the revised version, I am satisfactory with the answers and changes provided by the authors. Therefore, I recommend the revised manuscript to be accepted.

Appendix A

Response to Reviewers' Comments on Manuscript No. RSOS-191592

Reviewer: 1

Comments to the Author(s): In this submitted manuscript, the authors described the esterification of palm oil industry by-product to produce biodiesel, and studied the properties of the resulted biodiesel. Some useful results were contained. In my opinion, this manuscript could be accepted after considering the following two points.

1) The stability of the used heterogeneous catalyst should be discussed. If possible, the properties of the synthesized catalyst and the recovered catalyst should be characterized.

Response: The stability of the used catalyst was investigated and have been discussed under section 4.1 and Table 2.

2) The advantage of the resulted biodiesel should be discussed. At least, a comparison in various properties should be provided between the synthesized biodiesel and some reported typical biodiesel.

Response: The advantages of the synthesized PFAD methyl ester has been discussed and compared to other typical biodiesel in Table 8 and discussed in section 4.4. of its sub-sections, 4.4.6.

Reviewer: 2

Comments to the Author(s): The manuscript describes the biodiesel production from palm oil industry by-products by acid-catalyzed esterification reactions. The reaction conditions were thoroughly optimized by the response surface methodology (RSM). The various properties of the produced fuels and the blends with petro-diesel were analyzed. It is recommended to be published after revisions.

1) In Table 1, the first column (variables) can be elongated to accommodate the whole word. Besides, the detailed reaction conditions can be provided as the footnote.

Response: The correction has been done and the detailed reaction variables have been provided in the text of manuscript.

2) In Table 2, the "Wt.%" should be changed into "values", because the unit of the values are not all by weight.

Response: The correction has been done and "wt.%" has been changed to "values" as suggested.

3) The FTIR spectra are shown in an unclear manner. The figure should be replotted with higher resolution and different blends should be denoted by various colours.

Response: The FTIR has been replotted and the different spectra denoted with different colors, as suggested

4) The recyclability of the heterogeneous catalysts should be examined.

Response: The reusability of the PWB-H₂SO₄ biobased heterogeneous catalyst has been discussed under section 4.1.